# A Study on Noise Reduction of Gear Pumps of Wheel Loaders Based on the ICA Model

**DOI:** 10.3390/ijerph16060999

**Published:** 2019-03-19

**Authors:** Panling Huang, Liang Xu, Chuan Luo, Jianchuan Zhang, Feng Chi, Qi Zhang, Jun Zhou

**Affiliations:** 1School of Mechanical Engineering, Shandong University, Jinan 250061, China; hfpl@sdu.edu.cn (P.H.); xl9931120@163.com (L.X.); luoc_oo@163.com (C.L.); zhoujun@sdu.edu.cn (J.Z.); 2Key Laboratory of High Efficiency and Clean Mechanical Manufacture, Shandong University, Ministry of Education, Jinan 250061, China; 3Shandong Lingong Construction Machinery Co., Ltd., Linyi 276023, China; feng.chi@sdlg.com.cn (F.C.); qi.zhang@sdlg.com.cn (Q.Z.)

**Keywords:** gear pump noise, construction machinery, noise reduction and ICA

## Abstract

In order to reduce the noise level of wheel loaders caused by gear pumps and realize environmentally sustainable development, the noise generation mechanism of a gear pump was studied, and the influence of flow pulsation and gear impact on noise was analyzed. In order to reduce the interference of other noise sources on the noise level of the gear pump, a noise test rig was established. The mixed noise signals obtained from the rig test were separated using the ICA model. The ICA model includes the following algorithms: The fast Fourier transform (FFT), independent component analysis (ICA) and inverse fast Fourier transform (IFFT). Some theories about the influence of the teeth number and teeth profile on noise were analyzed by theory and simulation. A noise reduction strategy was proposed by increasing the teeth numbers and modifying the teeth profile of the gear pump. The tests results showed that the noise values of the external and the driver’s ear of the wheel loader were reduced to 1 and 2.2 dB (A), respectively. This proves the effectiveness of the optimization scheme of gear pump noise reduction.

## 1. Introduction

After entering the 21st century, in order to achieve environmental sustainability, environmental protection technology and information technology of construction machinery are put forward, which makes construction machinery enter a new stage of development. European and American markets have put forward stricter requirements for noise control of construction machinery products. Gear pumps are the units that perform the energy conversion from the mechanical energy to the pressure energy of oil. They are extensively used in construction machineries due to their advantages such as small size, light weight, insensitivity to oil pollution, reliable operation, good performance, and low production costs [1,2]. However, their significant disadvantage is relatively high noise emission [3]. Construction noise can cause a series of detrimental health effects on human beings, such as noise-induced hearing loss [4], arrhythmia, dyslipidemia, cancer, thyroid gland disorders and psychic disorders, and so on [5]. Therefore, the construction noise level must be reduced in order to develop environmental sustainability and improve the market competitiveness of construction machinery.

The causes of noise generation are as follows [6]: (1) Pressure shock and cavitation arising from the trapped volume between meshing teeth; (2) the noise caused by some manufacturing errors of gears during gear meshing; (3) flow pulsation and pressure pulsation caused by the volume change of the suction chamber and discharge chamber. 

In addition to sustainable development needs, reducing noise levels can also increase the life and reliability of equipment. At present, many scholars are studying the noise generation mechanism and noise reduction strategy of gear pump. Many studies have shown that pressure and flow pulsations are the main causes of noise generation, not the noise radiation of the pump itself. The intrinsic noise of gear pumps mainly depends on the vibrations and dynamic load on the gears. These dynamic loads are transmitted through the bearings in pump housing and generate noise [7]. Mucchi [8] thought that the noise levels are the consequence of the dynamic forces within the gear pump system, related with the flow and pressure ripple as well as the variable meshing stiffness and gear errors. Zhao [9] believed that in the design of the involute tooth profile of the traditional external gear pump, the inherent cause of significant flow inhomogeneity leads to undesired noise emissions and mechanical vibrations for involute teeth design of the traditional external gear pump. Therefore, continuous-contact helical gear pumps were proposed and successfully commercialized to replace the traditional gear pumps in the recent past. However, the high manufacturing cost of helical gear restricts its popularization and application in construction machinery. Some scholars have also carried out research work on noise reduction of piston pumps [10,11].

Independent component analysis (ICA) is a technique recently used in data analysis and signal processing. It is used in many fields: Blind source separation [12], image restoration of blurred images [13], brain magnetic resonance image analysis [14], telecommunications [15], financial data analysis [16], facial identification [17], and so on. In this paper, ICA methods were used in the signal processing of gear pumps. The purpose of using ICA signal separation is to verify the effectiveness of gear pumps before and after optimization.

In order to avoid the deviation of gear pump noise caused by other components in the whole machine noise test, the following two effective measures were adopted in this paper: The noise test rig of the gear pump was set up to avoid the influence of other noise sources, and the noise signals of gear pumps were separated by using the ICA model. The noise generation mechanism of the gear pump was analyzed by combining theoretical analysis with the finite element analysis (FEA) method. A noise reduction strategy of gear pumps was put forward based on the above analysis. Finally, noise tests were carried out to verify the effectiveness of noise reduction. 

## 2. Research Foundation and Content

Figure 1 shows the main research contents of this paper. In early research, the values of the external radiation noise and the driver’s position noise were obtained by the test of the whole machine noise for the wheel loader. The test and sound pressure calculation of wheel loaders were carried out in accordance with the requirements of national standards GB/T25614-2010 and GB/T25612-2010 of the People’s Republic of China. The sound pressure values of the wheel loader are shown in Table 1. Among them, CE (the abbreviation of French, it means EUROPEAN CONFORMITY in English) certification values were obtained according to European CE certification requirements. It can be seen from the table that the measured noise value was greater than the CE certification value, so it is necessary to study the noise reduction of the wheel loader. Through spectrum analysis and the acoustic array noise location method, it was found that the gear pump and the engine were the main noise sources of wheel loaders. Therefore, the next study will be divided into two parts. This paper mainly studies the noise reduction theory of the gear pump. The theory of engine noise reduction will be developed in the next stage. 

## 3. Methodology

The gear pump studied is shown in Figure 2. With the meshing transmission between the driving wheel and the driven wheel, the meshing volume between the teeth also changes. When the meshing teeth on the inlet side are separated gradually, the volume of the inlet port increases and the pressure decreases, and the liquid in the suction pipe is sucked into the pump. It is then carried around the sides of the gears by the teeth. Last, the fluid in the pump is delivered to the output port and then into the working oil circuit. Then, new fluid is sucked from the inlet port and discharged with the meshing of the gears. Through these operations, the oil suction and oil discharge of the gear pump in the hydraulic system can be completed, so as to provide power for the hydraulic system.

### 3.1. The Noise Generation Mechanism of the Gear Pump

There are many factors that cause the noise generation of gear pumps. However, considering the manufacturing cost, and not changing the volume structure of the gear pump, this paper will mainly consider the influence of flow pulsation and meshing impact on noise generation. 

#### 3.1.1. The Flow Pulsation

Usually, gear meshing makes a significant contribution to pressure and flow ripples. Ripples are main sources of vibrations, noise, and efficiency loss of gear machineries [18]. The instantaneous flow rate of gear pumps is uneven and varies with time. However, when the hydraulic pump rotates continuously, the instantaneous flow varies, according to the same rule. This phenomenon is called the flow pulsation of the gear pump. The flow pulsation of the gear pump determines the performance of the hydraulic system. If the flow pulsation of a gear pump is large, it will not only make the stability and uniformity of the working element worse, but also cause the vibration of the whole hydraulic system and emit a higher decibel noise. Therefore, for noise reduction measures of gear pumps, we should first consider reducing the flow pulsation.

The flow pulsation formula is expressed as follows [19]:(1)$$\delta =\frac{Q_{\max}-Q_{\min}}{Q}=\frac{Q_{\max}-Q_{\min}}{i\cdot n\cdot q}$$where *Q*_max_ and *Q*_min_ represent the maximum and minimum instantaneous flow, *Q* represents the average flow rate of the gear pump, *i* represents the speed drive ratio between the gear pump and motor, *n* represents the speed of the motor, and *q* represents the output volume of the gear pump.

By further calculation, the flow pulsation of the gear pump with side gap meshing is as follows:(2)$$\delta =\frac{3\pi^{2}\cos^{2}\alpha_{0}}{12（z+1)-\pi^{2}\cos^{2}\alpha_{0}}$$ and the flow pulsation of the gear pump without side gap meshing is as follows:(3)$$\delta =\frac{3\pi^{2}\cos^{2}\alpha_{0}}{48（z+1)-\pi^{2}\cos^{2}\alpha_{0}}$$where *α*_0_ represents the pressure angle of the meshing gears and *z* represents the teeth number of the gears.

From the above formula, it can be seen that increasing the pressure angle and the teeth number of the gear can reduce the pulsating flow.

#### 3.1.2. The Meshing Impact of Gear Teeth

Besides the flow pulsation of gear pumps, the meshing impact of gear teeth also produces noise due to the meshing characteristics of gears. As shown in Figure 3, for a pair of gears, the base pitch of the driving gear (Pb1) is different from that of the driven gear (Pb2) due to the profile error, installation error and bearing deformation during meshing transmission. Different base pitch can cause change of the instantaneous transmission ratio when it enters or exits meshing, which causes vibration and noise.

The noise reduction measures of the gear meshing impact should start with reducing impact energy and making gear meshing more stable. The main noise reduction method in this paper was to improve the gear processing accuracy by profile modification. Drum profile modification is the most widely used method. In most cases, the drum shape under the centrosymmetric condition can effectively compensate for the unbalanced load of gears, except for the excessive skewness or the small load. The factors to be considered in drum profile modification include drum shape and drum center position.

Generally, the teeth number of gear pumps is between 6 and 30. In the application of the loader, the requirement of displacement uniformity of gear pumps is not high, but there are certain requirements for installation size of a gear pump and the radial force on the gear to ensure the service life of the bearing and gear pump.

### 3.2. The Noise Reduction Strategy

Noise is emitted from the sound source, radiated to the outside, and then transmitted to the receiver through a certain sound transmission way. According to the mechanism of noise generation, noise can be controlled through three aspects: Noise source, the sound transmission path, and the receiver. So, the following noise reduction strategies can be adopted [20].

#### 3.2.1. Reducing Noise at the Sound Source

It is most fundamental and effective to control noise directly at the sound source. The following methods can be used to reduce noise: Developing and selecting low noise equipment, optimizing production and processing technology, selecting the appropriate materials and structures, or improving the machining and assembly precision of parts.

In this paper, noise reduction of the hydraulic gear pump was mainly achieved by increasing the teeth number of the gear pump and modifying the profile of gears. This is a method to reduce the noise of the sound source. 

#### 3.2.2. Control Noise in the Transmission Path

When the pressure of technical and economic cost exists in noise source control, the transmission path becomes the preferred measure. Usually, in construction machinery, sound insulation, sound absorption, vibration damping, and noise elimination are often adopted to achieve noise control in the transmission path. 

In the previous study, some sponges were pasted onto noise sources for noise reduction (as shown in Figure 4). This is a noise control method from the transmission path.

#### 3.2.3. Control Noise at the Receiver 

Operators can be protected by wearing earplugs, and equipment and instruments can be protected by sound insulation and vibration isolation. However, in engineering machinery, it is difficult for drivers to take corresponding protective measures. 

### 3.3. Noise Measurement of the Test Rig

The measurement and calculation methods refer to the national standard (GB/T 3767-1996), “Acoustics—Determination of sound power levels of noise sources using sound pressure—Engineering method in an essentially free field over a reflecting plane” [21]. Figure 5 shows a schematic diagram of noise measurement of a gear pump. The gear pump is driven by a motor (model: SIEMENS 180kW DC (Siemens, Munich, Bayern, Germany)), and the motor speed is measured by an optical speed sensor (model: PZ-V31P, Keyence, Japan). Noise signals are measured by six acoustic sensors (model: PCB-378B02, GRAS, Denmark). Multi-channel data acquisition instrument (model: scada02, LMS, Belgium) and a Laptop are used to collect and analyze the signals.

The test rig for noise measurement is shown in Figure 6, where (a) represents the layout of acoustic sensors (2a = 0.9 m, 2b = 1.18 m, c = 1.12 m), and (b) represents the test site. In order to reduce the influence of motor noise on gear pump noise, a partition wall is used between the motor and gear pump.

During the test, the motor speed was set to 2200 rpm, which is the same as the rated speed of a gear pump when it works normally. At the same time, the relief valve was adjusted to make the normal working pressure 20 MPa. Under this pressure, the radiation sound pressure of the test pump was measured. The original normal gear pump was selected in the test.

By calculating the sound power of the gear pump, it was concluded that the acoustic power of the sampling pump (2200 r/min, working pressure 20 MPa, and oil temperature 50 °C) was 100.6 dB (A). Adjust the relief valve to normal working pressure of 20 MPa. 

### 3.4. Noise Signal Separation Based on the ICA Model

In this test, the noise signal obtained includes not only the noise of the gear pump, but also the noise of the motor and other accessories. The ICA algorithm is used to separate the desired noise signal from the mixed noise. 

The transmission process of the gear pump is complex, and there is a time delay, a path effect, and reflective reverberation in the propagation process. Therefore, the traditional time domain of the ICA algorithm is not suitable for the separation of noise signals. In order to solve this problem, this paper extended the separation algorithm to the frequency domain.

#### 3.4.1. The ICA Model

Suppose there are *n* unknown independent statistical sources, which are transmitted by unknown channels and eventually accepted by *m* receivers. Considering the influence of the path delay and reverberation in signal transmission, the received signal of the receiver (*x_i_(t)*) is no longer a simple linear instantaneous mixing but a linear convolution mixing:(4)$$x_{i}(t)=\sum_{j=1}^{n}\sum_{p=0}^{P}h_{ij}(p)s_{j}(t-p)$$where *h_ij_* is the response function between the *j*th source signal and the *i*th receiver, *s_j_(t − p)*, is the delay function of the *j*th source signal to the *i*th receiver through different channels. The model is a convolution mixed model expressed as:(5)X(t)=H(t)∗S(t)where *X*(*t*) = [*x*_1_(*t*), *x*_2_(*t*), …*x*_m_(*t*)]^T^ is the mixed matrix obtained by the receivers, *S*(*t*) = [*S*_1_(*t*), *S*_2_(*t*), … *S*_m_(*t*)]^T^ is the source signal matrix, and *H*(t) is the mixture matrix in the time domain.

At present, the ICA algorithm based on the linear instantaneous mixed model is not suitable for convolutional mixing. Therefore, this paper used Fourier transform to transform the convolution mixing problem from the time domain to the frequency domain. The formula is as follows: (6)X(f)=H(f)S(f)where *X*(*f*) = [*x*_1_(*f*), *x*_2_(*f*), … *x*_m_(*f*)]^T^ is the observed signal matrix in the frequency domain, *H(f)* is the *m*×*n* hybrid matrix, and *S*(*f*) = [*s*_1_(*f*), *s*_2_(*f*), … *s*_n_(*f*)]^T^ is the source signal matrix. In the process of signal separation, we need to find a linear separation matrix *W* to obtain the estimated signal from the source signal. Hence, the separation of ICA can be expressed as:(7)Y(f)=WX(f)=W[H(f)S(f)]where *Y*(*f*) is the estimation of signal sources in the frequency domain. 

The separation process of the mixed signals based on the ICA model is shown in Figure 7.

The specific description is as follows: The mixed signals are collected by a data acquisition instrument and converted into the frequency domain by fast Fourier transform (FFT). The ICA algorithm is used to separate mixed frequency domain signals into various source signals. Fast Fourier inverse transformation (IFFT) is used to transform the frequency domain signals into time domain signals.

#### 3.4.2. Spectrum Analysis

The noise data of six measurement points are analyzed by 1/3 octave, the frequency spectrum and the weighted average. Figure 8 shows the average value of the sound pressure spectrum, which provides a theoretical basis for noise source analysis.

The maximum peak frequency in the figure is close to 330 Hz, which is consistent with the meshing frequency of the gear pump (calculated according to Formula (8), the teeth number z is 9). The second is the larger peak frequency of 36.7 Hz, which is the basic frequency of the motor (calculated according to Formula (9)). Other prominent frequency components are integral multiple frequencies of the fundamental frequency of the gear pump: (8)$$f=\frac{n}{60}\cdot z=\frac{2200}{60}\cdot 9=330\;\mathrm{Hz}$$where *n* is the motor speed, and *z* is the teeth number of the gear pump (9)$$f=\frac{n}{60}=\frac{2200}{60}=36.7\;\mathrm{Hz}$$

#### 3.4.3. Noise Extraction from the Mixed Noise

In this paper, the ICA model was used to deal with the noise data of six measurement points. The noise signal of the gear pump was extracted to obtain a more accurate measurement value, and the noise level of the gear pump before and after optimization were compared.

The time-domain signals of six measuring points were processed by fast Fourier transform to obtain frequency-domain signals, and then the frequency-domain signals were pretreated by means processing, whitening, the eigenvalue decomposition method, and so on. It was estimated that there were three independent sources. Figure 9 is the spectrum of three source signals separated by the ICA model.

As can be seen from the figure, the main peak frequency of signal 1 was 1982 Hz, and the amplitude was only 0.0852. The frequencies were mainly concentrated in the middle and high frequency bands. It was estimated that the noise may be caused by accessories such as relief valves. The main peak frequency of signal 2 was 330.3 Hz (meshing frequency of the gear pump). The amplitude of signal 2 was obviously larger than that of the other two signals, including medium and high frequencies such as 1321 Hz (4 × 330.3, four order) and 1652 (5 × 330.3, five order), and so on. Therefore, it can be inferred that the main noise comes from gear pump. 

The corresponding frequency of signal 3 was 36.5 Hz, which is consistent with the motor speed, and the amplitude was about 0.23. Therefore, it can be deduced that the noise comes from motor. 

IFFT was used to transform signal 2 in the time domain to get the noise signal of the gear pump. The final sound pressure was 87.6 dB (A) by calculation.

## 4. Gear Pump Optimization Strategy and Test Verification

### 4.1. Gear Pump Optimization Strategy

This paper combined the noise generation mechanism of the gear pump and reduced the gear pump noise by increasing the teeth number and improving the tooth profile of the gear pump.

#### 4.1.1. Increasing the Teeth Number of the Gear Pump

(1) The flow pulsation

According to Formula (2), the flow pulsation of a gear pump is related to the pressure angle and the teeth number of the gear pump. The teeth number of the original gear pump was 9, and the pressure angle was 20°. Considering the cost of gear processing and strength, the maximum teeth number was set to 12 in this paper. In accordance with Formula (2), the influence of different teeth numbers on the flow pulsation is shown in Table 2.

It can be seen from the table that the flow pulsation of the 12-tooth was 5.7% less than that of the original 9-tooth gear pump under the constant volume structure of the gear pump. 

(2) The meshing force

Generally speaking, under the same working conditions, for the same system, the smaller the meshing force, the smaller the impact noise. Therefore, the meshing force between gears is analyzed by using finite element analysis software (Hyperwork). The rigid body element is used to simulate the intermediate shaft body, and the 5 freedom degrees of the rigid element are restrained to release the freedom degree of rotation. The displacement of the gear pump is 166 milliliter per revolution, the motor speed is set to 2200 r/min, and the system pressure of the gear pump is set to 20 MPa. Hence, the output torque *M_t_* is calculated by Formula (10): (10)$$M_{t}=\frac{Pqv}{\mu 2\pi }$$where *P* is the outlet pressure of the gear pump, *q_v_* is the displacement of the pump (mL/rev), and *μ* is the mechanical efficiency of the pump, set to 0.9. Therefore, *M_t_* calculated by Formula (10) is 587 N·m. 

Under the above conditions, the meshing transmission of the gear pump was simulated by software. Figure 10 shows the input conditions and the simulation results of Mises stresses at different times, where (a) and (b) are the motor speed and output torque loaded on the gear pump, respectively, and (c) and (d) are the Mises stress nephogram of the original 9-tooth and the 12-tooth gear pump corresponding to a certain time.

The meshing forces of gears at different times were calculated according to the change of the stress value and contact area between gear teeth at different times, as shown in Figure 11. As can be seen from the figure, the meshing force of the 12-tooth gear pump was obviously lower than that of the 9-tooth gear pump, and the maximum meshing force was reduced by about 30% at a time.

#### 4.1.2. Gear Profile Modification

From the above analysis, gear profile modification can be used to improve machining accuracy, thereby reducing noise. In this paper, the method of drum shape modification was adopted. Gear manufacturing accuracy can control the influence of gear surface roughness and tooth profile errors on noise. Therefore, the two involute gears of the gear pump were repaired, and the requirements are shown in Figure 12.

where DNA is the addendum circle, D_y_ is the minimum trim diameter, D_w_ is the pitch diameter, and DLNF is the dedendum circle. The tooth profile error is controlled by the left picture, and B is the tooth width. Drum shaped machining can prevent tooth surface contact.

Five gear shafts were repaired with a forming gear grinder, and then the tooth profile was detected with a tooth profile detector. A set of gear shafts was selected; their tooth profile is shown in Figure 13, where *y*-axis represents the radial direction of the gear. The picture on the left is before repair, and that on the right is after repair. Each lattice represents 0.01 mm in the picture. By measurement, the shape deviation between the tooth profile and the tooth orientation was between 0.01~0.02 mm, which satisfied the requirement. 

### 4.2. Optimized Gear Pump Test Verification

The noise value of the optimized gear pump was tested on the test rig, and then the ICA model was used to get the noise signal of the gear pump in the time domain. Finally, the noise test of the wheel loader equipped with the optimized gear pump was carried out to verify the optimization effect of the improved gear pump. Table 3 shows a noise values comparison of the gear pump before and after optimization. 

From the table, the sound power radiation value of the optimized gear pump obtained on the test rig was 1.7 dB (A), lower than that of the original one. The external radiation noise was reduced by 1 dB (A), and the driver’s position noise was reduced by 2.2 dB (A) through an external radiant noise test. Combined with Table 2, the driver’s position noise value of the optimized pump was less than the CE authentication values (80 dB (A)), but the external radiation noise value was greater than the CE authentication values (107 dB (A)). Therefore, in the next stage, the muffler structure of the engine will be redesigned to reduce the external radiation noise.

## 5. Conclusions

In this paper, noise measurement and the structure optimization of a gear pump were carried out in order to reduce the noise value of gear pumps of wheel loaders. The noise test rig was built to reduce the influence of noise from other parts on the noise of the gear pump. The ICA model was established to separate gear pump noise from mixed noise. By increasing the teeth number of the gear pump, the flow pulsation of the gear pump under the rated condition was reduced by 5.7%. The meshing force of 12 teeth was obviously lower than that of 9 teeth and reduced by about 30% by simulation analysis. Finally, the noise test verified the effectiveness of the noise reduction method by increasing the number of gear teeth and profile modification. The test results showed that the noise value of the gear pump was reduced by 1.7dB (A) after optimization, the external radiation noise was reduced by 1 dB (A), and the driver’s position noise was reduced by 2.2 dB (A). The noise reduction theory can be extended to other construction machinery fields.

## Figures and Tables

**Figure 1 ijerph-16-00999-f001:**
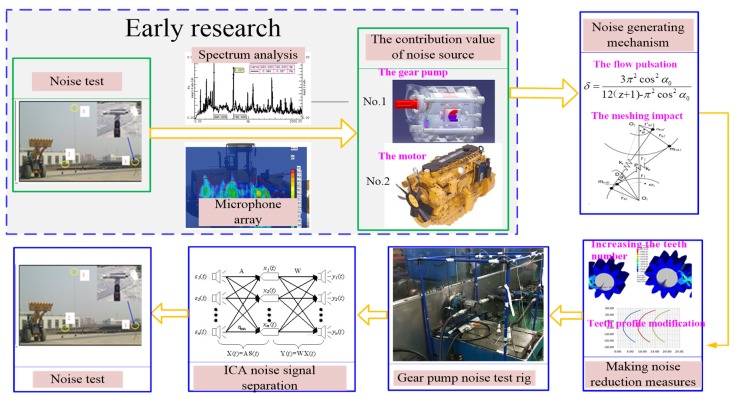
The main research contents.

**Figure 2 ijerph-16-00999-f002:**
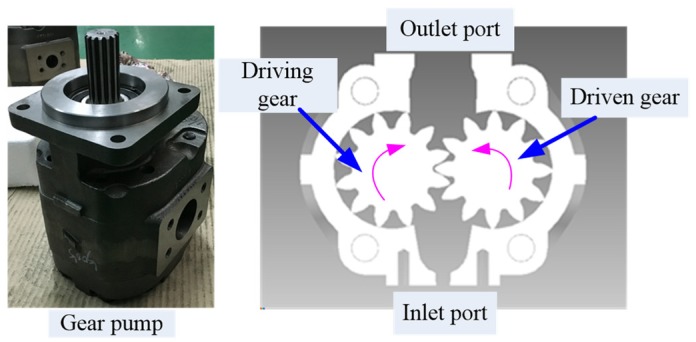
Gear pump structure diagram.

**Figure 3 ijerph-16-00999-f003:**
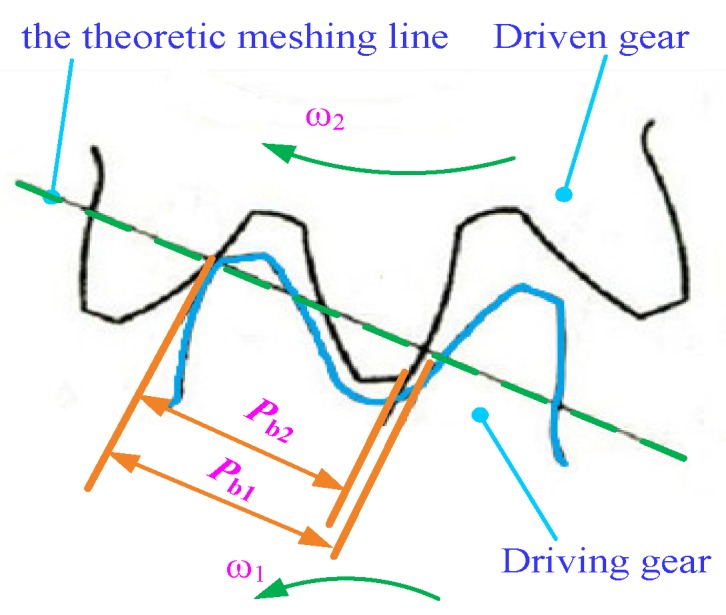
Mechanism of meshing impact.

**Figure 4 ijerph-16-00999-f004:**
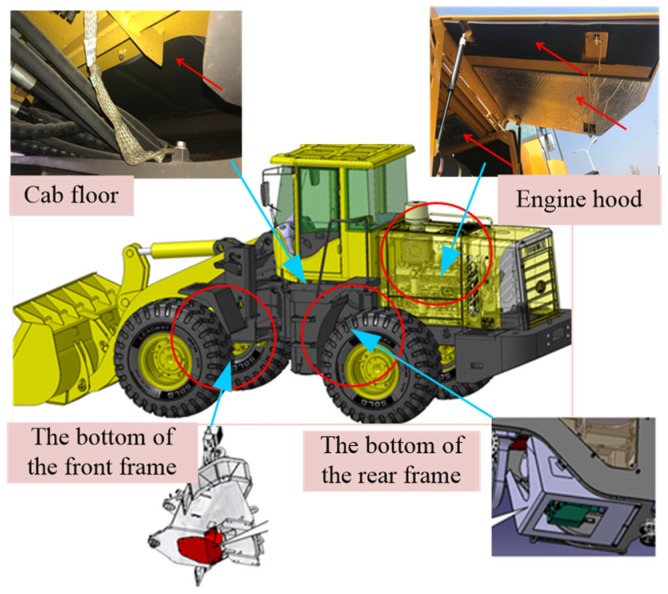
Pasting sponge to reduce noise.

**Figure 5 ijerph-16-00999-f005:**
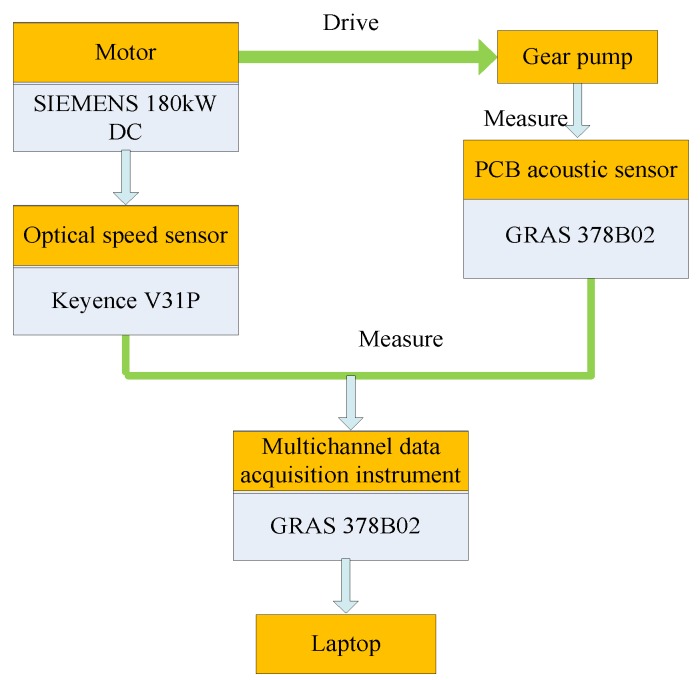
Schematic diagram of noise measurement.

**Figure 6 ijerph-16-00999-f006:**
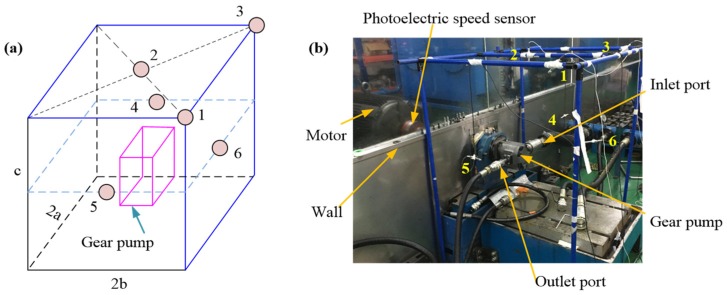
Test rig used for noise measurement. (**a**) Acoustic sensor layout, (**b**) test site.

**Figure 7 ijerph-16-00999-f007:**
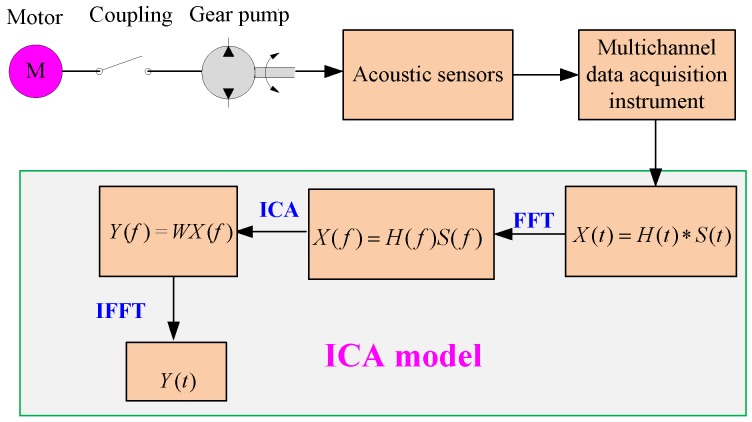
Separation process of the noise source of the gear pump.

**Figure 8 ijerph-16-00999-f008:**
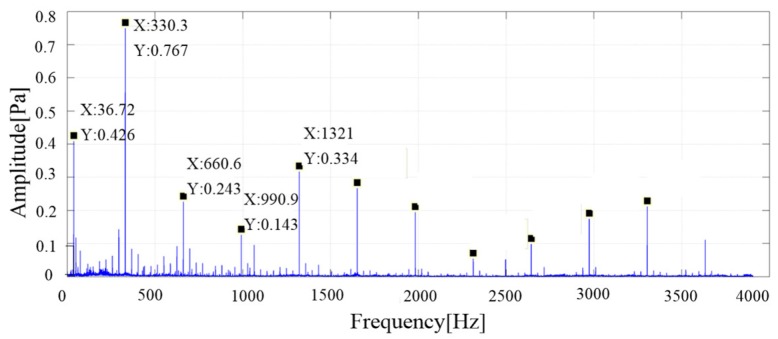
Noise pressure spectrum of the original gear pump test rig.

**Figure 9 ijerph-16-00999-f009:**
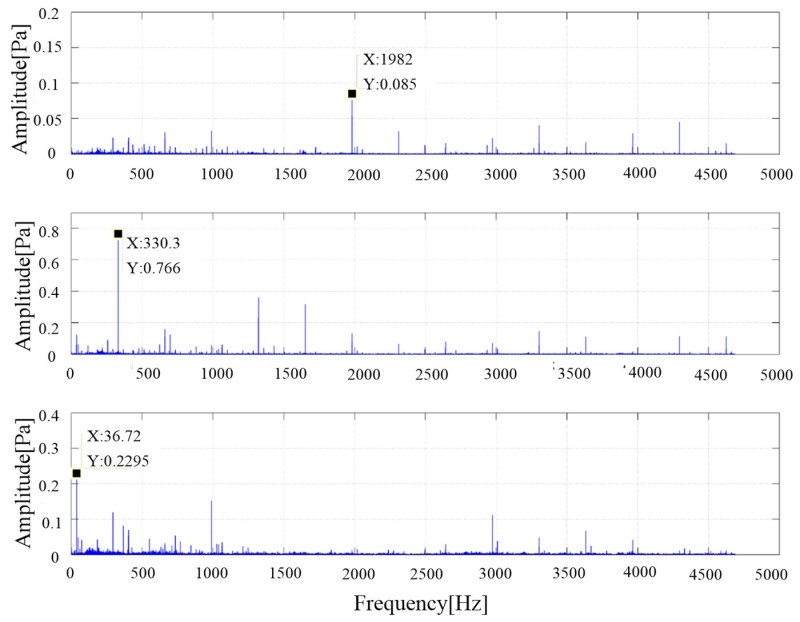
Results of noise separation of the test rig of the original gear pump.

**Figure 10 ijerph-16-00999-f010:**
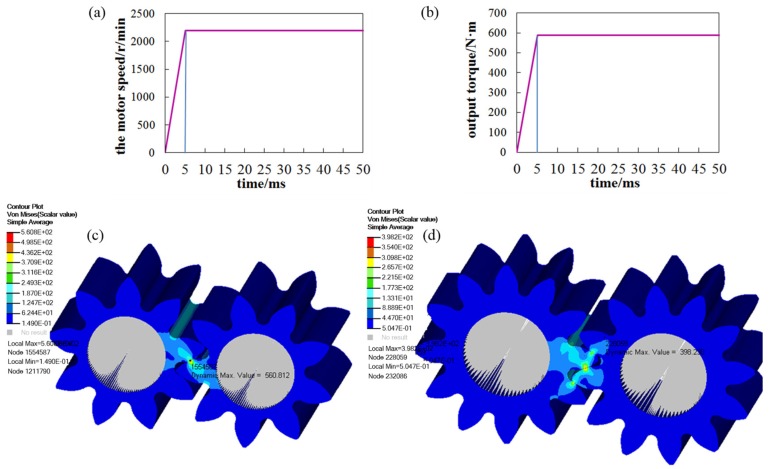
The meshing forces of the gear pump.

**Figure 11 ijerph-16-00999-f011:**
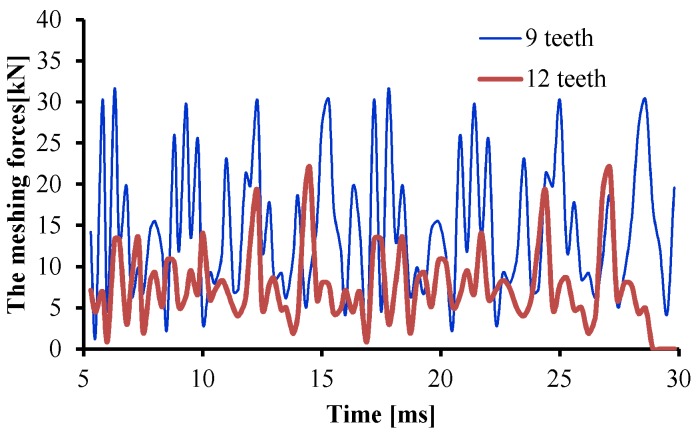
The meshing forces of the gear pump.

**Figure 12 ijerph-16-00999-f012:**
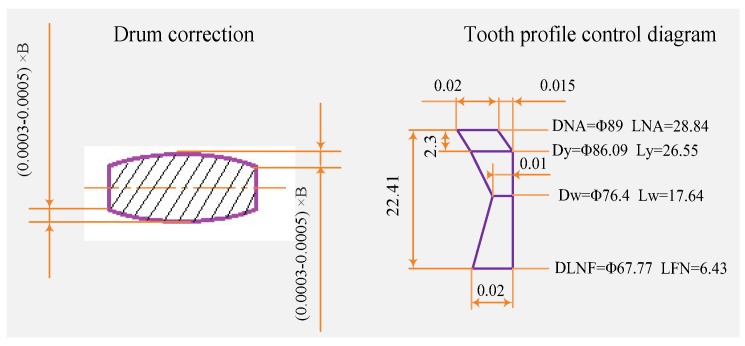
Gear repair requirements.

**Figure 13 ijerph-16-00999-f013:**
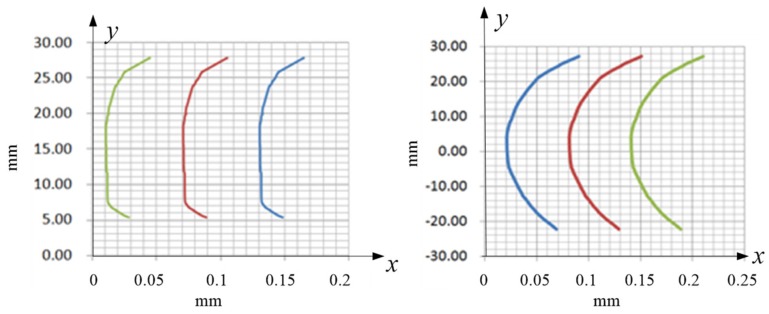
Detection results of the teeth profile modification.

**Table 1 ijerph-16-00999-t001:** The noise value of the wheel loader.

	Measured Values dB(A)	CE Certification Values dB(A)
The driver’s position noise	81.5	<80
The external radiation noise	110.3	≤107

**Table 2 ijerph-16-00999-t002:** The flow pulsation with different teeth numbers.

The Teeth Number *z*	Flow Pulsation Rate *δ* (%)
9	23.5
10	21.2
11	19.3
12	17.8

**Table 3 ijerph-16-00999-t003:** Noise values comparison of the gear pump before and after optimization.

Symbol	Original Pump dB (A)	Optimized Pump dB (A)	Reduction Value dB (A)
The mixed noise value	100.6	95.9	4.7
The gear pump noise	87.6	85.9	1.7
The driver’s position noise	81.5	79.3	2.2
The external radiation noise	110.3	109.3	1

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
