# Peer review of "A Study on Noise Reduction of Gear Pumps of Wheel Loaders Based on the ICA Model"

_ijerph, 2019, doi:10.3390/ijerph16060999_

Round 1

Reviewer 1 Report

Dear authors,

in the following you can find some comments\suggestions:

- please check the positions of the equations numbers;

- please check the text for minor errors (e.g. line 339);

- the legend in Fig, 10c and Fig. 10d is not clear;

- please add labels to the axes in Fig 13

Thank you 

Best regards

Reviewer 2 Report

1) Please add 1-3 sentences about future research

2) The authors should cite new references (2016-2019 Web of Science). 

for example 30 new references.

Please show that you have new knowledge about acoustics

for example

New efficient two channel forward set-membership partial-update NLMS algorithms for blind speech enhancement and acoustic noise reduction

APPLIED ACOUSTICS

Volume: 141  Pages: 322-332

DOI: 10.1016/j.apacoust.2018.07.020

Published: DEC 1 2018

Noise Reduction Method of Underwater Acoustic Signals Based on Uniform Phase Empirical Mode Decomposition, Amplitude-Aware Permutation Entropy, and 

Pearson Correlation Coefficient

ENTROPY

Volume: 20  Issue: 12

Article Number: 918

DOI: 10.3390/e20120918

Published: DEC 2018

Acoustic-Based Fault Diagnosis of Commutator Motor

ELECTRONICS

Volume: 7  Issue: 11

Article Number: 299

DOI: 10.3390/electronics7110299

Published: NOV 2018

Bearing fault diagnosis of a permanent magnet synchronous motor via a fast and online order analysis method in an embedded system

MECHANICAL SYSTEMS AND SIGNAL PROCESSING

Volume: 113  Pages: 36-49  Special Issue: SI

DOI: 10.1016/j.ymssp.2017.02.046

Published: DEC 2018

Recognition of acoustic signals of commutator motors 

APPLIED SCIENCES

8 (12), 2630, 2018. 

DOI: 10.3390/app8122630

Acoustic emission-based condition monitoring methods: Review and application for low speed slew bearing

MECHANICAL SYSTEMS AND SIGNAL PROCESSING

Volume: 72-73  Pages: 134-159

DOI: 10.1016/j.ymssp.2015.10.020

Published: MAY 2016

Fault detection of electric impact drills and coffee grinders using acoustic signals

SENSORS 

19 (2), 269, 

DOI: 10.3390/s19020269

Published: 2019

Effect of layouts of wells and the walls of wells on noise reduction effect

APPLIED ACOUSTICS

Volume: 145  Pages: 228-233

DOI: 10.1016/j.apacoust.2018.08.019

Published: FEB 2019

Condition monitoring and fault diagnosis of motor bearings using undersampled vibration signals from a wireless sensor network

JOURNAL OF SOUND AND VIBRATION

Volume: 414  Pages: 81-96

DOI: 10.1016/j.jsv.2017.11.007

Published: FEB 3 2018

Reviewer 3 Report

The manuscript deals with the implementation of strategies to reduce the noise level generated by a gear pump and on the assessment of the new noise level by a methodology based on a fast Fourier transform, independent component analysis and an inverse fast Fourier transform, built to separate the mixed noise signal of the combined motor-gear pump system. The manuscript is of interest to implement noise reduction strategies from the specific case of gear pumps. In my opinion, the authors should stress the importance of their paper for potential applications to other sectors, at least by emphasizing the applicability of their methodology based on ICA in other contexts. These aspects should be included in the conclusions, which must be strengthened and not limited to a mere summary of the paper. Finally, English level is poor in some parts and requires revision. Please also consider the following comments.

Specific comments:

- Page 1, Line 14: The term “sustainability development of the environment” sounds quite inappropriate; it could be replaced by “sustainable occupational environments” for instance.

- Page 1, Line 35: The cited reference does not mention gear pumps; the authors should replace it with another relevant reference.

- Page 2, Line 64: Please provide at least one reference to support this sentence.

- Page 2, Lines 65-76: Please clearly state the novelty of the manuscript.

- Page 2, Line 78: The sentence “The driving gear… to rotate” should be corrected, since its meaning is not clear.

- Page 3, Lines 89-96: Please provide a reference for your previous study. If not published, please add a few details on the measurement methodology at least.

- Page 3-4, Lines 96-108: This part should be moved to the introduction of the manuscript.

- Page 5, Line 171: Do you mean the teeth number of the gear pump or the number of gear pumps? Please specify.

- Page 5, Lines 179-180 and Figure 4: This part should be removed from the manuscript since it is not particularly relevant.

- Page 6, Line 208: Please re-phrase this sentence.

- Page 9, formula 8: I suppose that the factor 9 is related to the number of teeth, as explained later. Please explain this in the correspondence of formula 8 instead.

- Page 10, Lines 289-290: This part should be removed as inessential to the manuscript.

- Page 10, Line 305: It is obvious that everything refer to this paper, so “in the paper” should be removed from the text.

- Page 10, Line 316: Please replace “shown in Fig. 10” with “as shown in Fig. 10”.

- Page 12, Line 335: This line seems the continuation of a sentence. Please correct this.

- Page 12, Lines 335-337: English requires revision.

- Page 12, Fig. 13: It is not clear what section of the teeth is represented in this figure. Please provide more clear explanations.

- Page 12, Line 361: Please replace “chapter” with “paper”.

- Pages 12-13, Lines 364-376: This part should be thoroughly re-written. The three points do not summarize the conclusion of the work but the work itself. The conclusions should focus on the importance of the findings of the paper for the field of study and on the future potential applications of the presented methodology in other cases.

Round 2

Reviewer 2 Report

--